# Aptamer-Based Sensor for Rapid and Sensitive Detection of Ofloxacin in Meat Products

**DOI:** 10.3390/s24061740

**Published:** 2024-03-07

**Authors:** Siyuan Wang, Xiuli Bao, Mingwang Liu, Qingfang Hao, Shuai Zhang, Siying Li, Lei Zhang, Xinxin Kang, Mingsheng Lyu, Shujun Wang

**Affiliations:** 1Jiangsu Key Laboratory of Marine Bioresources and Environment/Jiangsu Key Laboratory of Marine Biotechnology, Jiangsu Ocean University, Lianyungang 222005, China; siyuanwang@jou.edu.cn (S.W.); xlbao@jou.edu.cn (X.B.); mwliu@jou.edu.cn (M.L.); qfhao@jou.edu.cn (Q.H.); 20222210128@jou.edu.cn (S.Z.); 2022220837@jou.edu.ac (S.L.); leizhang@jou.edu.cn (L.Z.); kangxinxin@jou.edu.cn (X.K.); mslyu@jou.edu.cn (M.L.); 2Co-Innovation Center of Jiangsu Marine Bio-industry Technology, Jiangsu Ocean University, Lianyungang 222005, China

**Keywords:** aptamer, ofloxacin, SELEX, biosensor, rapid detection

## Abstract

Ofloxacin (OFL) is widely used in animal husbandry and aquaculture due to its low price and broad spectrum of bacterial inhibition, etc. However, it is difficult to degrade and is retained in animal-derived food products, which are hazardous to human health. In this study, a simple and efficient method was developed for the detection of OFL residues in meat products. OFL coupled with amino magnetic beads by an amination reaction was used as a stationary phase. Aptamer AWO-06, which showed high affinity and specificity for OFL, was screened using the exponential enrichment (SELEX) technique. A fluorescent biosensor was developed by using AWO-06 as a probe and graphene oxide (GO) as a quencher. The OFL detection results could be obtained within 6 min. The linear range was observed in the range of 10–300 nM of the OFL concentration, and the limit of the detection of the sensor was 0.61 nM. Furthermore, the biosensor was stored at room temperature for more than 2 months, and its performance did not change. The developed biosensor in this study is easy to operate and rapid in response, and it is suitable for on-site detection. This study provided a novel method for the detection of OFL residues in meat products.

## 1. Introduction

Fluoroquinolone (FLQ) is a synthetic antimicrobial drug, which is widely used in the treatment of gastrointestinal and respiratory tract infections in animal husbandry and aquaculture. FLQ offers several advantages, such as strong antimicrobial activity, broad antimicrobial spectrum, high bioavailability, and low price [1,2]. Ofloxacin (OFL) is a third-generation synthetic FLQ, which prevents DNA replication by inhibiting DNA topoisomerases and DNA deconjugating enzymes and has inhibitory effects on both Gram-negative and Gram-positive bacteria [3]. The slow degradation of OFL and its long residual time in the environment and high activity result in a variety of adverse reactions in animals and even human beings when utilized inappropriately [4]. Although some studies have reported the use of aptamers for the detection of OFL, most of the aptamer sequences used were screened by Reinemann et al. The libraries used for the screening of aptamers were more complex, and the sequences were longer [4]. For example, label-free fluorescent detection using an SYBR Green I (SG-I) fluorescent insert combined with ofloxacin aptamer was linear in the range of 1.1–200 nM, with a detection limit of 0.34 Nm [5]. On the other hand, the electrochemical detection of OFL in tap water and wastewater treatment plants was performed using a sensor composed of OFL aptamer and gold nanoparticles with a detection range of 5 × 10^−8^ × 10^−5^ M and detection limit of 1 × 10^−9^ M [6]. Similarly, the aptamer-based photoelectrochemical detection of OFL was attempted using polydopamine-Ag 2S nanocomplexes sensitized with TiO_2_ nanotube arrays, which was linear in the range of 5.0 pM–100 nM with a detection limit of 0.75 pM [7]. The library used for screening aptamer in this study was simpler and more affinitive, and the developed biosensor had a wider linear range and lower detection limit.

Currently, the commonly used detection methods for antibiotics are electrochemical detection [8,9,10,11], microbiological detection [12,13,14], and liquid chromatography–mass spectrometry [15]. Although these detection methods are highly sensitive and rapid, they are complicated, require expensive instruments and specialized operators, and are not suitable for on-site detection. Therefore, there is a need to develop an efficient and sensitive method that can be used for on-site detections of antibiotics. In recent years, biosensors have emerged as the detectors used in medical, food, and other analytical fields to achieve a rapid, sensitive, and on-site monitoring of antibiotics [16,17]. A biosensor consists of two parts: a biorecognition element (enzyme, aptamer, antibody, synthetic molecularly imprinted polymer (MIP), microorganism, etc.) and a signaling element (electrical signal, fluorescence, etc.) [18]. Compared with traditional detection methods, biosensors are highly sensitive, highly specific, rapid, cost-effective, easy to use, and portable [19]. Recently, electrochemical sensors, fluorescent sensors, surface plasma sensors, etc., have been used for the detection of antibiotic residues of animal origin [20,21,22].

Aptamer is a segment of ssDNA that shows high affinity and specificity for a target, which is SELEX based on artificially constructed oligonucleotide libraries [23,24,25,26,27]. As recognition progenitors, biosensors bind to targets, thereby transforming from free conformations to city-specific three-dimensional structures, such as hairpin structures, stem loops, and G-quadruplexes [28,29,30,31]. Biosensors bind specifically to the target through hydrogen bonds, hydrophobic interactions, van der Waals forces, etc., and generate recognition signals through signal conversion elements [32]. Aptamers are cost-effective and show high affinity, good specificity, wide target range, and good stability [33,34].

The objective of this study was to screen the aptamers that show high sensitivity and specificity to OFL. Magnetic bead SELEX technology was used to design and develop an aptamer-based fluorescent biosensor for OFL detection using an EP tube as the carrier. When OFL was added to the sensor, the sensor emitted a blue fluorescent signal. This study provides a new strategy for screening aptamers of other antibiotics and a method for detecting OFL residues in meat products.

## 2. Materials and Methods

### 2.1. Material

The library and primers used in this experiment were synthesized by Sangon Biotech (Shanghai, China), while OFL, enrofloxacin (ENR), norfloxacin (NOR), and benzoxacillin (OXA) were purchased from Shanghai Yuanye Biotechnology. Chloramphenicol (CPL), sulfadiazine (SD), roxithromycin (CCR), penicillin G (PG), tobramycin (TOB), tris (hydroxymethyl)-aminomethane (Tris), urea, low molecular weight DNA Marker, taq PCR premix, and 4S red plus nucleic acid stain were purchased from Shanghai Sangong Biotech (Shanghai, China). Amino magnetic beads, N, N-dimethyl formamide (DMF), 1-Hydroxybenzotriazole (Hobt), bromo-trispyrrol-idinophosphonium-hexafluophosphate (PyBroP), ampicillin (APR), NaCl, MgCl_2_, and N-(2-hydroxyethyl) piperazine N′-(2-ethane sulfonic acid) (HEPES) were purchased from Aladdin Biochemical Technology for the in vitro screening of aptamers. Agarose gel DNA recovery kit was purchased from Tiangen Biochemical Technology (Beijing, China). Agarose was purchased from Biowest. All solutions were prepared using ultrapure water with a resistance of >18.20 MΩ, which was obtained using Bamstead Labtower EDI water purification system (Thermo, Dreieich, Germany).

### 2.2. Methods

#### 2.2.1. Immobilization of OFL on Amino Magnetic Beads

As shown in Figure 1A, carboxyl groups were observed in the structure of OFL. In this experiment, the amidation reaction between the amino and carboxyl groups resulted in the formation of an amide bond to couple OFL with aminated magnetic beads. After the addition of ssDNA, the screening process was accomplished by applying a magnetic field. OFL was mixed with DMF, Pybrop, Hobt, and DIPEA and activated at 9000× *g* for 1 h under ambient conditions. The aminated magnetic beads were blown through DMF (650 μL) 3 times and then resuspended in 100 μL of DMF. The activated OFL was incubated overnight with the resuspended magnetic beads at 9000× *g* (room temperature) to form a complex. After magnetic separation, the beads were washed twice with 650 μL of DMF and blown twice through screening buffer (650 μL). The beads were then resuspended in 100 μL of screening buffer (×2) and stored at 4 °C. Since OFL contains “π-π” conjugated groups with fluorescent properties, the coupling rate of OFL with amino magnetic beads was determined by measuring the fluorescence intensity at excitation and emission wavelengths of 290 nm and 800 nm, respectively, before and after coupling, using the multifunctional enzyme labeling instrument (Appendix A).

#### 2.2.2. Aptamer Screening

The full length of the random ssDNA library used in this experiment was 75 bp. It contained a primer-binding region of 20 nucleotides on both sides and a random region of 35 nucleotides in the middle. The random library sequence was 5′TACATCACCTAATCCTGCGG-N35-GATTGGGTCATTACCGAGGA-3′, while forward and reverse primer sequences were 5′-TACATCACCTAATCCTGCGG and 5′-TCCTCGGTAATGACCCAATC-3′, respectively. Prior to screening, random libraries and primers were purified and recovered using 15% denaturing polyacrylamide gel (dPAGE). The DNA sequences were amplified and enriched by PCR. After PCR, amplicons were detected and recovered, and the DNA concentration was determined by using 3% agarose gel. The recovery rate was calculated, and the amplicons were used for the next round of screening. The PCR reaction mix (50 μL) consisted of 0.5 μL of the template DNA, 0.5 μL of each primer, and 25 μL of Taq PCR Mix. The PCR cycling conditions were as follows: pre-denaturation at 95 °C for 3 min, 20 thermal cycles (denaturation at 95 °C for 15 s; annealing at 54 °C for 15 s; and extension at 72 °C for 1 min), and final extension at 72 °C for 5 min. The PCR products were stored at 4 °C.

The empty magnetic beads, OFL-coated magnetic beads, and competitive magnetic beads were washed three times with a screening buffer before each round of screening, and the supernatant was removed after magnetic separation. According to the recovery rate, the libraries were diluted to 100 μL using the screening buffer before use and then denatured at 95 °C for 10 min. Then, the libraries were immediately placed on ice for 10 min and then taken out and placed at room temperature to form its structure. The screening process is shown in Figure 1B. Negative screening was performed, and the ssDNA was incubated with empty and non-competitive magnetic beads (ENR, NOR) at room temperature and at 9000× *g* for 1 h. The inactive DNA was removed by magnetic separation, retaining the supernatant. Then, the positive screening of active DNA was carried out by incubating the supernatant with magnetic beads coated with OFL. After screening, the magnetic beads were washed twice with 650 μL of screening buffer (×2) to remove inactive DNA. Subsequently, 650 μL of ultrapure water was used to wash twice again. Then, 100 μL of elution buffer was added for denaturation at 95 °C for 5 min. The supernatant was collected, and the procedure was repeated four times. Finally, ssDNA was recovered by alcohol precipitation, and PCR amplification was performed to complete the screening process.

#### 2.2.3. Sequencing and Analysis

After the screening, the PCR products were sent to Sangon Biotech for high-throughput sequencing. The sequencing data were used to predict the secondary structure of aptamers using the IDT OligoAnalyzer 3.1 software (https://sg.idtdna.com/UNAFold (accessed on 1 August 2023)), and representative aptamers were selected and synthesized based on the secondary structure, GC content and binding energy (ΔG), and labeled with FAM tag at its 5′ end.

#### 2.2.4. OFL Measurement

The fluorescence of labeled aptamer was mixed with graphene oxide (GO), and the GO could adsorb ssDNA on the surface by “π-π” stacking [35]. GO is a fluorescence resonance energy transfer (FRET) acceptor, which can quench fluorescent groups on the nucleic acid aptamer [2]. When ofloxacin was added, the aptamer was stripped from GO and the fluorescence was restored. The concentration of GO was optimized. GO concentrations were set as 0.1, 0.5, 1, 5, 10, 20, 30, and 50 μg/mL. Then, we added 10 μL of 1 μM candidate aptamers to 20 μL of GO at different concentrations and mix thoroughly. Then, seventy microliters of 1 μM OFL were added and mixed well. The fluorescence intensity was measured.

Each experiment was carried out in a light-avoiding 96-well plate, with a total volume of 100 μL. To ensure the accuracy and validity of the results, each experiment was repeated three times. The fluorescence intensity of each sample was measured using a multifunctional microplate detector, at excitation and emission wavelengths of 488 nm and 535 nm, respectively. The fluorescence intensities of experimental group (F) and control group (F0) were recorded, and ΔF (ΔF = F0 − F) was calculated to determine the fluorescence intensity of OFL under different experimental conditions.

#### 2.2.5. Affinity and Specificity of Aptamers

To determine the binding affinity of aptamers to OFL, candidate aptamers were diluted to the final concentrations of 0.5, 2.5, 5, 7.5, 15, 20, 25, and 30 nM under the same conditions using buffer. Next, 10 μL of candidate aptamer was added to 20 μL of GO and mixed well. Then, 70 μL of OFL was added, and the fluorescence intensity was measured to calculate ΔF. A nonlinear fitting curve was plotted using Origin 2022, and the dissociation constant (Kd) of each candidate aptamer was calculated according to the following formula: Y = Bmax × X/(Kd + X).

Kd is an important indicator to assess the affinity of aptamers to OFL, with smaller Kd representing greater affinity [36]. Y is the average value of fluorescence of aptamer bound with OFL. X is the concentration of aptamer, and Bmax is the maximum binding capacity of aptamer.

To determine the specificity of aptamers for OFL, fluorescence intensities of seven antibiotics (CCR, OXA, CPL, SD, PG, TOB, and APR) were determined. Equal amounts of OFL and antibiotics were added to the well plates, and equal amounts of ultrapure water were used in the control group. The real fluorescence intensity was determined by using a multi-kinetic enzyme marker, and the fluorescence phenomenon was observed under a blue light illuminator (Safe Imager^TM^). Finally, the specificity of the candidate aptamers for OFL was determined by calculating ΔF and observing the fluorescence coloring images.

#### 2.2.6. Optimization of Aptamer Reaction Conditions

To maximize the accuracy and sensitivity of the fluorescence quenching method for the determination of OFL, aptamer concentration and reaction time were optimized. Three parallel experiments were performed for each combination of reaction conditions.

For the optimization of aptamer concentration, aptamer solutions with final concentrations of 5, 10, 20, 50, 75, 100, 150, and 200 nM were taken into a light-avoiding 96-well plate, and then the 1 μM of OFL was added in the wells. The total volume of the reaction system was 100 μL. The fluorescence intensity of each sample was measured, and the ΔF was calculated to determine the optimal aptamer concentration.

To optimize the reaction time of aptamer and OFL, the aptamer was allowed to react with OFL for 0, 2, 4, 6, 8 and 10 min, respectively, at the optimal concentrations of aptamer and GO. Then, the optimal reaction time with stable fluorescence was determined by measuring the fluorescence intensity.

#### 2.2.7. Preparation of the Aptamer Sensor

As shown in Figure 2A, the biosensor was prepared in an EP tube (200 μL) using aptamer WO-06 and GO. After adding 500 nM ofloxacin and mixing well, the fluorescence intensity was measured using a multifunctional enzyme marker, and the sensor was proven to be viable if green fluorescence appeared under the irradiation of blue light.

#### 2.2.8. Characteristics of Aptamer Sensor

The OFL detection performance of the sensor was evaluated by determining its sensitivity, specificity, and stability. Three parallel experiments were conducted for each group.

To determine the sensitivity of the sensor, OFL was diluted using ultrapure water to final concentrations of 5, 10, 50, 100, 150, 200, 300, 400, 500, 600, and 700 nM. OFL was added to the sensor, and ultrapure water was added to the control group. The total volume of the system was 100 μL. The reaction solution was mixed thoroughly for 6 min, and then its fluorescence intensity was measured. Finally, ΔF was calculated, and a nonlinear fitting curve was plotted to obtain a linear graph. The limit of detection (LOD) was calculated according to 3σ/S, where “σ” is the standard deviation and “S” is the slope of curve.

The specificity of the sensor was determined by using non-target antibiotics. The concentration of 1 μM of OFL and other antibiotics were added in the sensor group, while equal amounts of ultrapure water were used in the control group. Real fluorescence intensity was determined while observing the fluorescence under a blue light illuminator. Finally, the specificity of the sensor for OFL was determined by calculating ΔF and by observing the fluorescence chromogenic images.

To study the stability of sensor as well as the effect of sugar preservatives on stability of the sensor, two kinds of sensors (with and without preservatives) were prepared. The prepared sensors were kept at room temperature for 0–3 months, and the fluorescence intensities of sensors were measured after the addition of OFL. The fluorescence was observed under a blue light illuminator.

#### 2.2.9. Testing of Actual Samples

Beef, pork, chicken, duck, and shrimp were purchased from the market and fully ground. Then, 10 g of meat products were weighed and added into 10 mL of wood spirit for extraction. The mixture was put into a biological sample homogenizer for homogenization and then centrifuged at 8000× *g* for 20 min. Supernatant was filtered through a 0.22 μm filter. The extracted sample was subjected to high performance liquid chromatography (HPLC) to detect the presence of OFL in the samples. According to GB 31650.1-2022 [37], the residue standard of OFL in meat is lower than 5.5 nM, which was used as the detecting concentration of OFL to spike samples. Additionally, 100 μL of the extract samples and spiked samples were detected the fluorescence, and 70 μL of the extract samples and spiked samples were added into the sensors. Then, the images of tubes fluorescence were taken, and the fluorescence intensity was measured. The meat with the lowest background signal was selected for the LOD experiments. The final concentrations of OFL were 5, 10, 50, 75, 100, 150, 200, 250, and 300 nM, and the fluorescence intensity was measured by adding the extract into the sensor to calculate the LOD.

## 3. Results and Discussion

### 3.1. Aptamer Screening, Sequencing, and Analysis

As shown in Appendix A, the OFL coupling rate was around 40%, which indicated that the amino magnetic beads were completely encapsulated, and the influence of other functional groups as well as the binding sites were reduced due to this supersaturation. A total 12 rounds of screening (9 rounds of forward screening and 3 rounds of reverse screening) were conducted using the OFL-coated amino beads as the stationary phase. The results revealed a decrease in the enrichment rate when reverse screening was introduced. This decrease can be attributed to the removal of inactive ssDNA by the increased number of screenings. The aptamers with low affinity to the stationary phase were eluted gradually, while the ones with high affinity were enriched. The enrichment rate gradually increased with the increased number of screenings, from 9.65% in the first round to 54.21% in the twelfth round of screening. Due to the dramatic increase in enrichment rate, the screening process was terminated at the twelfth round, and the screening products were sent for high-throughput sequencing. As shown in Appendix A, the screening products were amplified by PCR, and the quality of amplicons was checked through 3% agarose gel electrophoresis. The clarity of the bands improved gradually, and their positions conformed to the library. This confirmed that the aptamer sequences obtained after screening had higher affinity to OFL.

Through high-throughput sequencing, a total of 169,603 aptamer sequences were obtained. The secondary structure as well as the predicted ΔG of the sequences with high enrichment rate were determined using IDT OligoAnalyzer 3.1 software, and the results are shown in Table 1. As shown in Figure 2B, the aptamer sequences were mainly composed of stem-loop and hairpin structures and had high GC content. The presence of these structures enabled the aptamer to specifically recognize OFL. Based on the secondary structures as well as ΔG, six sequences were selected as the candidate aptamers and named AWO-01, AWO-02, AWO-03, AWO-04, AWO-05, and AWO-06. These aptamers were labeled with an FAM tag at their 5′ ends and synthesized.

### 3.2. Characterization of Aptamers

As shown in Figure 2B, the fluorescence signal gradually became weaker with the increase in GO concentration. The fluorescence intensity remained stable at GO concentration of 30 μg/mL. Therefore, the GO concentration used in the subsequent experiments was 30 μg/mL.

The respect affinities of the candidate aptamers to OFL were determined as described in Section 2.2.5. Kd values of the six candidate aptamers were estimated by measuring the fluorescence intensity and then plotting a nonlinear fitting curve (Appendix A and Figure 2C). Figure 2C shows that AWO-06 had the lowest Kd (28.16 nM). It indicates that AWO-06 had the strongest affinity for OFL.

Furthermore, the selection specificity of the candidate aptamer AWO-06 was analyzed by using CCR, OXA, CPL, SD, PG, TOB, and APR as the controls for OFL, while ultrapure water was used as the blank. The results showed that the candidate aptamer AWO-06 produced a strong fluorescence signal for OFL, showing its selective specificity for OFL (Figure 2D). 

At present, the electrolytic dissociation constants derived from the present study have an advantage over the Kd = 130.1 nM reported by Yuhong Zhang et al. [35] for ofloxacin and Kd = 0.11–56.9 nM reported by Reinemann et al. [33]. In conclusion, AWO-06 has a high affinity and specificity for OFL. Therefore, the aptamer was selected for subsequent experiments.

### 3.3. Optimization of Reaction Conditions

In order to achieve the highest fluorescence signal, two key factors were optimized, including aptamer concentration and reaction time. As shown in Figure 3A, the aptamer concentration was first optimized by selecting the aptamer concentration range of 5–200 nM. ΔF was determined to find a balance between the generation of an obvious signal and a lower background signal. With the increase in aptamer concentration, the fluorescence signal intensity gradually increased. However, when the aptamer concentration was too high, the effect of GO on fluorescence quenching was poor, which further caused an increase in the background signal. When the aptamer concentration was 75 nM, the fluorescence signal intensity was strong, and the background signal was low. In addition, the reaction time (within 0–10 min) was optimized at the aptamer concentration of 75 nM. Furthermore, the reaction time (within 0–10 min) was optimized at an aptamer concentration of 75 nM. As shown in Figure 3B, the reaction occurred instantly after adding the OFL, producing an obvious fluorescence signal. With the increase in reaction time, the fluorescence signal gradually enhanced and reached equilibrium at 6 min. Therefore, the optimal reaction time was 6 min. In summary, the optimal concentration of the aptamer was 75 nM, while the optimal reaction time was 6 min.

### 3.4. Optimization of Reaction Conditions for the Developed Sensor

When OFL was added to the sensor, a strong fluorescent signal was produced by the sensor. On the other hand, no fluorescent signal was observed in the absence of OFL, which proved that the sensor was feasible (Figure 4A). The performance of the sensor (i.e., sensitivity, specificity, and stability under optimal conditions) was examined to determine whether it can be effectively used for OFL detection. 

Sensitivity is an important index to analyze the performance of the sensor. In the present study, an OFL concentration of 5–700 nM was used to analyze the sensitivity of the sensor. As shown in Figure 4B, the fluorescence intensity of the sensor gradually increased with the increase in the concentration of OFL. Furthermore, the nonlinear fitting curve revealed that there was a good linear relationship between OFL and AWO-06 in the range of 10–300 nM, with a detection limit of 0.61 nM. Subsequently, the specificity of the sensor was determined by measuring the fluorescence intensity of the sensor to calculate the ΔF and by observing the fluorescence image of the sensor under blue light irradiation (Figure 4B). As shown in the colorimetric image (Figure 4C), the sensor was highly specific for OFL, while the signals for other antibiotics were relatively low or even non-existent. Finally, the stability of the sensor was determined (Figure 4D). Firstly, the effect of sugar preservative on the stability of the sensor was analyzed. The fluorescence intensity graph as well as the fluorescence color rendering image revealed that the sensor with the sugar preservatives was more stable and better to recognize OFL. In the second step, the performance of the sensor was examined after storing the sensor at room temperature for 0–3 months. The sensor was observed to be most effective in the first month. However, good fluorescence response was observed even after 3 months of storage. It proved that the developed sensor can be stored for a long time.

In previous reports, several methods have been investigated for the detection of OFL. Most of these methods are based on HPLC, liquid chromatography–mass spectrometry (LC-MS), etc. K. Elaslani et al. detected OFL by coupling nanomaterials with other instruments (such as a pulsed electrochemical detector) using silver particle-modified carbon paste electrode, and the study reported a linear range between 4.0 × 10^−6^ and 1.0 × 10^−3^ M and an LOD of 9.47 × 10^−7^ M [38]. Similarly, Yan Wu et al. investigated a fluorescent probe based on a Zn/Eu-MOF (OFL) assay with a linear range between 0.1 and 80 μM and an LOD of 0.44 μM [2]. These studies suggest that the linear ranges of these methods are generally narrow, and the operation is complicated, which is not conducive to rapid on-site testing. Compared to these reported methods, the method presented in this study had a wide linear range of 10 to 300 nM, with an LOD of 0.61 nM. Moreover, the present method was easy to operate and faster and did not need expensive instruments, complex pre-treatment processes, and professional operators.

### 3.5. Detection of OFL in the Meat Products

The presence or absence of OFL in the meat extract was first determined by HPCL. The results showed that none of the 12 selected meat products contained OFL (Appendix A). Then, the sample extract and the standardized extract were added into the sensor, and the fluorescence intensity was measured. The fluorescence color images were taken after mixing the solution. As shown in Figure 5A, most of the meat products had a high background effect. Based on this observation, Beef D was selected to determine the LOD of the sensor, which was 8.76 nM (Figure 5B). The results of this experiment revealed that the background signal of the extract was high. This suggested that the sample contained some interfering substances, which affected the results. However, the spiking experiment showed that this sensor can be used for the detection of OFL in meat products.

## 4. Conclusions

In this study, aptamers obtained from the ssDNA library were screened using magnetic bead SELEX technology, which could effectively and rapidly detect OFL. An aptamer biosensor was developed for the rapid and sensitive detection of OFL using the fluorescence quenching method based on GO. GO can effectively adsorb the aptamer and quench its fluorescence. Moreover, OFL can be bound to the aptamer, which may lead to structural changes and peel off from GO, causing changes in the fluorescence signal. Using this method, AWO-06 was found to be the best candidate aptamer, with a Kd of 28.16 nM. The specificity of aptamer was analyzed using other antibiotics, which proved that the aptamer was highly specific for OFL. Furthermore, AWO-06 was used to quantify OFL in the range of 10–300 nM with a detection limit of 0.61 nM. The designed sensor showed good sensitivity and specificity for OFL, and the OFL could be detected after 6 min of reaction between the sample and aptamer. The sensor also showed good detection performance in actual meat samples, with an LOD of 8.76 nM. The aptamer sensor could be stored for a long time at room temperature, showing good stability after 3 months of storage. The biosensor has the advantages of easy operation, rapid response, low cost, and no requirement for professional operators, etc. The developed sensor can rapidly detect the residues of OFL in meat products, and it has a great application potential in animal husbandry and aquaculture industry.

## Figures and Tables

**Figure 1 sensors-24-01740-f001:**
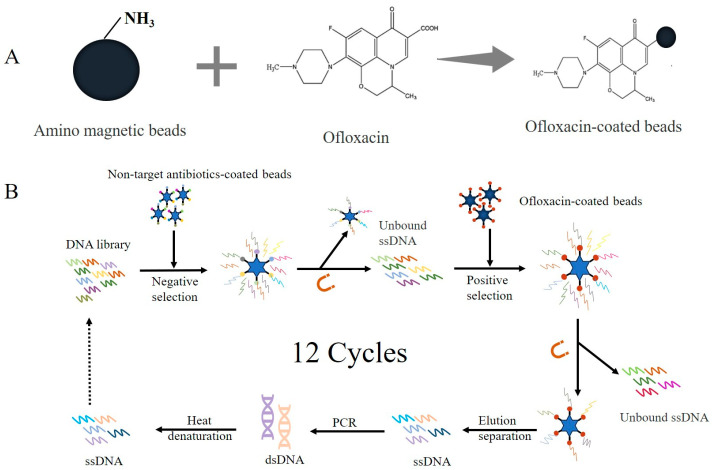
Aptamer screening: (**A**) coupling OFL to amino magnetic beads by amidation reaction; (**B**) screening process of aptamers of OFL.

**Figure 2 sensors-24-01740-f002:**
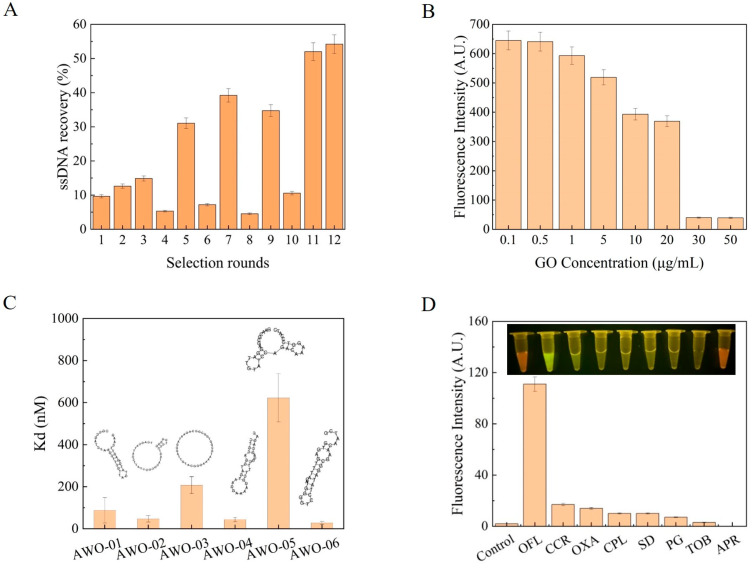
(**A**) Recovery rates during aptamer screening; (**B**) optimization of GO concentration; (**C**) Kd and secondary structure prediction map of candidate aptamer; (**D**) AWO-06 specificity assay. Inset: fluorescence chromatography images.

**Figure 3 sensors-24-01740-f003:**
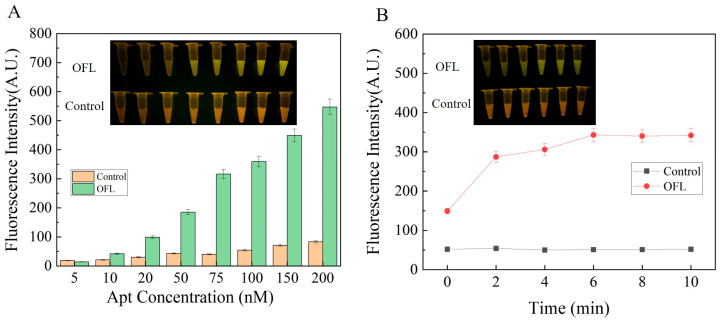
Optimization of reaction conditions: (**A**) optimization of aptamer concentration; (**B**) optimization of reaction time. Inset: fluorescence chromatography images.

**Figure 4 sensors-24-01740-f004:**
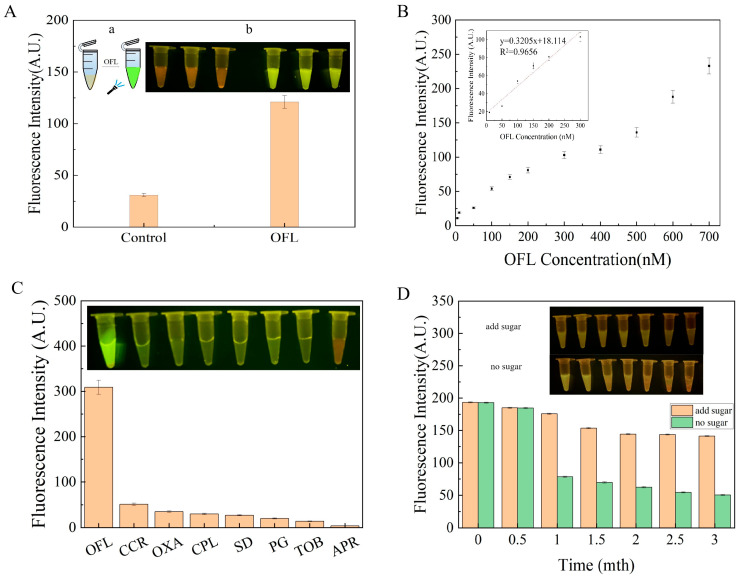
(**A**) Feasibility analysis of sensor: (a) sensor design, (b) fluorescence colorimetric images; (**B**) sensor sensitivity; (**C**) sensor specificity; (**D**) sensor stability. Inset: fluorescence colorimetric images.

**Figure 5 sensors-24-01740-f005:**
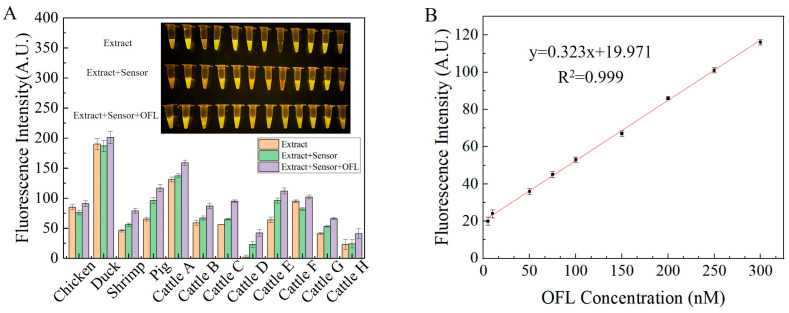
Detection of OFL in actual meat samples: (**A**) OFL detection in meat, inset: fluorescence colorimetric image; (**B**) LOD of aptamer-based sensor for Beef D.

**Table 1 sensors-24-01740-t001:** Sequences of candidate aptamers.

Name	DNA Sequences (5′-3′)	ΔG (kcal/mol)
Apt-WO-01	GAACTTGAGTGTATGATTGCTTCGAATCTAGCCGC	−3.60
Apt-WO-02	GGGGGTCCCTGATACGCGGTTATTCGATTACGACT	−4.54
Apt-WO-03	CAGAATAGGGATTGTACGGGAAAATGCGGTGGCGG	−2.78
Apt-WO-04	CAGACGAGGCTTTCGGAAAAGGAATGATGGACGTC	−4.29
Apt-WO-05	GAACGCGAGGATTGTATCCAGACAAAGGTCCCATC	−4.41
Apt-WO-06	GTGAGTTTACATGGGGTCCTATAGGCGAACAATCG	−4.80

## Data Availability

All the data are included within this article and the Appendix A, and they are available from the corresponding author upon request.

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
