# Peer review of "Aptamer-Based Sensor for Rapid and Sensitive Detection of Ofloxacin in Meat Products"

_sensors, 2024, doi:10.3390/s24061740_

Round 1

Reviewer 1 Report

Comments and Suggestions for Authors

SELEX is a widely used aptamer screening technology that can target a wide variety of molecules. Using this powerful technology, an ofloxacin sensor for the detection of meat products is developed in the paper. The workload of the study is relatively large and the data is relatively complete. It can be published in Biosensors after some revisions.

Comments: 

1 The paper mainly focused on aptamer screening and performance testing of the standard ofloxacin. On the other hand, the results showed that the background signal of meat samples was high, experiments and analysis of meat samples were relatively less, so I suggested that "in meat products" should be removed from the title.

2. SELEX in lines 16, 17 and 68 should be written in a standard and consistent way.

3. Primer-binding region of 20 nucleotides on either side of six aptamers are identical, unimportant and should not be listed in Table 1.

4. Insets of Figure 4. B should be improved. The characters are too small to read and Chinese cannot be displayed..

5. Really Supplementary Materials were not found on the web. They may not have been uploaded correctly.

Reviewer 2 Report

Comments and Suggestions for Authors

This work Wang et al developed an aptamer-based sensor through SELEX to detect Ofloxacin in various samples. The authors evolved aptamers to detect OFC by enriching them over a stationary phase which consisted of the Ofloxacin conjugated to magnetic beads. The enriched sequences were then modified to develop a fluorescent sensor with Graphene oxide acting as the quencher. The authors demonstrate a high dynamic range and sensitivity of their sensing system with stability at room temperature. I find the work of merit and significance and believe it should be accepted after these comments are addressed:

1)    Could the authors provide details on how they optimized the Graphene Oxide and Aptamer concentrations and how the various combinations of concentrations impact the sensitivity of the sensor

2)    Figure 2A, y-axis, units are missing, please add them

3)    Have the authors tested the performance of their sensor over a longer time to see how stable the signal is for prolonged periods of time, and if so, how could that be leveraged for use as a sensor

4)    How do the authors explain figure 2A, the loss in concentration after some rounds of screening for the aptamer.

5)    Have the authors looked into whether long term exposure could lead to quenching or bleaching of their fluorescent signal. How stable is the fluorescence itself.

6)    It would be great if the authors compared their technique directly with state of the art methods used to detect such compounds and compared the relative Kds, sensitivities and LOD to establish in greater detail the key benefits presented by their work.

I would also recommend to thoroughly read the manuscript as there are a few grammatical errors.

Comments on the Quality of English Language

Please look into some minor grammatical errors.

Reviewer 3 Report

Comments and Suggestions for Authors

The aptamer-based sensor used for rapid on-site detection of ofloxacin. A fluorescent biosensor was developed by using AWO-06 as a probe 17 and graphene oxide (GO) as a quencher. A fluorescent sensor designed for ofloxacin detection with the range of 10-300 nM of OFL concentration Limit of detection 0.13 nM. The sensor applied for direct detection of ofloxacin without complex pre-treatment.

The experimental part needs some corrections and more details. The conclusions are consistent with the evidence and arguments presented. The references are appropriate

The manuscript accepted after major revision

My questions to the authors are the following:

1-     In supplementary and no published files, the authors write highlights. the supplementary file should contain data of experimental work. 

2-    In figure 5A what is the concentrations of spiked OFL which used in meat samples.

3-    In Fig. 5 B the error bar of the calibration curve does not clear or added ?

4-    In Line 274 , you write (Fig. S3) , where it is ?

5-      In section which beginning with line 312,

A)   In a paragraph about figure 4A , write  the concentration of OFL which used in experimental work.

B)   authors write” As shown in Fig. 4C, the fluorescence intensity of the sensor gradually increased  with the increase in the concentration of OFL”  I think you mean Fig. 4B , please correct it

C)   The same for the specificity of the sensor should be figure  4C . for specificity study what is the concentration of OFL and interfering materials which used in experimental wok?

D)    For Fig 4C and 4D ,In Y axis ,  the fluorescence intensity of OFL is around 300 A.U. where in the calibration curve Fig 4B the maximum fluorescence intensity was 50 A.U. , explain how ?

6-      In section which beginning with line 330 , you illustrate the previous work , could you add the data in tables containing the materials which used , techniques, linear range and limit of detection to  compare it with your results. 

7-    Line 344  change HPCL word  with HPLC

8-    Line 354 change sone word  with some

9-    Line 354 , authors write “This suggested that the sample contained sone interfering substances which  affected the results” do you think about  some treatments before measurements and what about the pH value of the samples , does it effect on the experimental work ?

10- In line 345 you write figure S3 , where it ? supplementary file does not containing any figures.

11- Line 239 , Fig. S1  where it is ?

12-  Write definitions for all abbreviation  like CCR, OXA, CPL, SD, PG, TOB and APR

Please see an attached file. 
